# A Transcutaneous Randomized Pulsed Radiofrequency Application for Spine Pain Conditions: A Case Series

**DOI:** 10.3390/jfmk10030242

**Published:** 2025-06-25

**Authors:** Daniel de Moraes Ferreira Jorge, Olav Rohof, Melina Brigato Ferreira Jorge, Alexandre Teixeira, Cezar Augusto de Oliveira, Pablo Sobreiro, Douglas Freitas Dos Santos, Stephany Cares Huber, Jose Fabio Santos Duarte Lana

**Affiliations:** 1Instituto Regenem de Medicina Avançada, São Paulo 04532-002, SP, Brazil; 2Regenerative Medicine, Orthoregen International Course, Indaiatuba 13334-170, SP, Brazil; pablosobreiro@gmail.com (P.S.); douglascoluna@hotmail.com (D.F.D.S.); ioc.orthoregen@gmail.com (S.C.H.); josefabiolana@gmail.com (J.F.S.D.L.); 3Pain Clinic, Orbis Medical Centre, 6166 GE Sittard-Geleen, The Netherlands; o.rohof@gmail.com; 4Instituto Sensus de Medicina, São Paulo 04532-002, SP, Brazil; melinarb@hotmail.com; 5Clinica de Dor, R. São João de Brito 610, 4100-455 Porto, Portugal; alteix@gmail.com; 6Orthopedics, ABC Oliveira Medical Clinic, São Paulo 03310-000, SP, Brazil; drcezaroliveira@gmail.com; 7Department of Orthopedics, Brazilian Institute of Regenerative Medicine (BIRM), Indaiatuba 13334-170, SP, Brazil; 8Medical School, Max Planck University Center (UniMAX), Indaiatuba 13343-060, SP, Brazil; 9Clinical Research, Anna Vitória Lana Institute (IAVL), Indaiatuba 13334-170, SP, Brazil; 10Medical School, Jaguariúna University Center (UniFAJ), Jaguariúna 13820-000, SP, Brazil

**Keywords:** transcutaneous pulsed radiofrequency, spine pain, disc herniation, non-invasive therapy

## Abstract

**Background:** Transcutaneous Randomized Pulsed Radiofrequency (TCPRF-STP) is a non-invasive therapeutic approach increasingly explored for managing spine-related pain, particularly in cases involving disc herniations and degenerative spine conditions. **Objectives:** To evaluate the use of transcutaneous PRF-STP in the treatment of spine pathologies and its evolution in short-term follow-up. **Methods:** This case series examines the outcomes of three patients treated with TCPRF-STP for varying spine pathologies, including lumbar and cervical disc herniations, lumbar stenosis, and radiculopathy. All patients had previously undergone conventional conservative therapies without a satisfactory improvement and were unwilling or unable to undergo invasive procedures. The treatment involved the application of electromagnetic fields through adhesive skin patches at targeted sites. Patients underwent three sessions of TCPRF-STP, with follow-up assessments evaluating pain and MRI. **Results:** Transcutaneous PRF-STP showed notable reductions in pain (VAS 0 in most cases), improvements in movement, and the restoration of normal daily activities. Follow-up MRI scans demonstrated positive structural changes in the treated discs. Although long-term recurrence occurred in one case, the patient remained active without functional limitations. **Conclusions:** Transcutaneous PRF-STP offers a promising, minimally invasive alternative for patients seeking to avoid surgery, though further studies with larger cohorts and longer follow-up periods are necessary to establish more robust evidence of its efficacy. This technique could become an important adjunct in managing chronic spinal pain conditions, offering patients an option with minimal risk and hospital demands.

## 1. Introduction

Pulsed radiofrequency (PRF, frequency 420,000 Hz) has been widely used in the treatment of chronic and acute pain conditions since its invention by Menno Sluijter in 1998. It has shown promising results, particularly when applied with specific needles or cannulas that have an active, non-insulated tip, allowing electromagnetic fields to be generated percutaneously. The energy is delivered in pulses, with distinct periods of silence between discharges. These parameters, including the voltage, frequency, and pulse width, are precisely controlled by a PRF generator connected to the cannulas or skin electrodes [1,2].

PRF was developed as a safer alternative to continuous RF, which often reaches higher temperatures. PRF typically maintains tissue temperatures between 39 and 42 °C, avoiding the risks of ablation associated with temperatures above 43 °C [3,4]. Numerous investigations, including case reports, clinical studies, and randomized controlled trials (RCTs) have demonstrated the positive effects of PRF not only on neural tissues but in many other areas as well.

The mechanism of action of PRF involves neuromodulation through the electromagnetic field generated between the dispersive (grounding) electrode and the active electrode (cannula), this field primarily targets C and A-delta fibers, which are involved in pain transmission [5]. PRF inhibits pain signaling, stimulates descending inhibitory pathways in the posterior horn of the spinal cord (specifically the serotoninergic and noradrenergic systems) [6], and exerts anti-inflammatory effects by modulating the microglial activity in the central nervous system [7].

Intra-articular PRF has been shown to deliver satisfactory results, even when the needle is positioned at a considerable distance from the afferent nerves [8]. These findings suggest that PRF may exert anti-inflammatory effects on immune cells, broadening its potential applications.

PRF’s anti-inflammatory actions are achieved by reducing the accumulation of pro-inflammatory cytokines, such as IL-1b, IL-6, TNF-alpha, and MMP-3, at the treatment site while promoting the release of anti-inflammatory cytokines, including IL-10 and IL-17 [8,9,10,11]. The core mechanism was further clarified through studies on PRF’s effects in a standard muscle injury model in rats, demonstrating that PRF’s electromagnetic field reduces oxidative stress by recombining radical pairs, as noted by Brasil et al. [12].

Posteriorly, Menno Sluijter and colleagues published a paper proposing that PRF may facilitate “functional restoration” by modulating inflammation and tissue healing [13].

This anti-inflammatory, redox-based effect of PRF enhances the cellular and articular environment, supporting chondral tissue health and neural regeneration [14]. Its applications are broad, and its combination with orthobiologic products, as described by Jorge et al. in 2022 [15], offers new therapeutic possibilities for challenging clinical cases such as chronic pain [16], advanced osteoarthritis, chronic tendinopathies, frozen shoulder, peripheral neuropathies, discopathies (with or without herniation), and the degeneration of the cervical, thoracic, and lumbar spine joints [2,16,17].

In spinal applications, PRF can target joints (through a medial branch block or facet denervation) or be applied near the dorsal root ganglion (DRG), activating anti-inflammatory pathways in the posterior horn of the spinal cord and the neural cellular microenvironment. Multiple studies have described these effects in detail [2,8,15,16,17], which may account for the significant clinical improvements seen with minimally invasive or transcutaneous PRF applications, with minimal risks. Intradiscal applications of PRF have also been reported [18], and these procedures can be performed in outpatient settings without the need for anesthesia.

A new update of the pulsed radiofrequency technique was also later developed by Menno Sluijter in partnership with Alexandre Teixeira, from Portugal, in 2016, offering a randomized way to deliver pulsed radiofrequency energy, called STP (Sluijter Teixeira Poisson).

Sluijter and Teixeira introduced this irregular burst type of pulsed RF with an irregular (Poisson-type) distribution of time between the pulses and pulse width. Irregular pulses might have an enhanced biological effect. Cells around the electrode might recognize regular pulses as “non-self” stimuli that could be ignored, contrary to the irregular burst-type PRF STP.

Nowadays other electrotherapy modalities, like spinal cord stimulation and TENS, also use irregular rhythms with better clinical results than regular and constant stimulation [19,20].

In PRF STP, random electromagnetic field actions would be able to promote pain inhibition by stimulating long-term depression (LTD) and down-regulate long-term potentiation (LTP), which would cause pain intensification [18]. The wide variance of stimulus delivery could be associated with the maintenance of LTD, as described by Migliore et al. in 1999 [21].

PRF can be performed in an outpatient setting and offers clinicians an alternative or complementary approach to oral drug therapy and intra-articular injections. Furthermore, it may serve as a valuable treatment option for patients who are either unfit or unwilling to undergo surgical intervention [22].

Building on the work of Nordenstrom (Karolinska Institute, Sweden) [23], who reported promising results with electrotherapy in patients with malignancies based on the hypothesis that the electrical resistivity of the blood vessel wall is significantly higher than that of the blood itself—thus allowing the vascular tree to propagate electric fields—Sluijter and Teixeira treated several patients with intravenous PRF and successfully published four case reports on this treatment [24].

Sumintra Rampersad (Radboud University, Nijmegen, The Netherlands) conducted a finite computer simulation to calculate the strength of the electric field generated by intravenous PRF. Her findings indicated that a field strength of 200 V/m was effective, with this field strength extending approximately 5 to 6 cm around the active tip of the electrode [25]. Notably, cells naturally communicate via electric fields ranging from 50 to 250 V/m, which falls within the physiological range [26].

After intravenous PRF was outdated, Sluijter and Teixeira introduced the transcutaneous PRF, whereby E fields of 50–250 V/m could be applied in two ways:Locally, on a body region (shoulder or knee joint, liver, etc.), by placing the skin electrodes over the inflamed area. Two positive RCT’s have been published by Taverner et al. [23,27]Systemically, by placing the electrodes over the axillary artery and the forearm, and all the immune cells in the blood pass through the PRF electric field and are exposed, having an effect in the whole body.

TCPRF, or Redox PRF, offers a low-risk treatment option for joints, peripheral nerves, and conditions related to oxidative stress, without relying on high temperature peaks. By using special adhesive plates on the skin, TCPRF generates an electromagnetic field that can target specific tissue compartments, such as joints or other affected areas. While the use of needles and cannulas in PRF has already been demonstrated as a safe and effective option, as evidenced by a comprehensive review of more than 200 articles presented by Vanneste et al. in 2017 [2], the transcutaneous application of PRF presents an even less invasive and risk-free alternative for patients where needle-based treatments may be a hindrance. Furthermore, TCPRF may serve as first line of treatment before considering more invasive procedures, so we applied this technique in some clinical conditions with very interesting results, which are presented below.

## 2. CASE 1: Lumbar Root Pain

Patient 1, a 45-year-old woman, was referred to our clinic in São Paulo/SP, Brazil, with complaints of low back pain radiating to the right lower limb (MID), accompanied by paresthesias and a worsening pain with exertion. She rated her pain as VAS 8, which significantly limited her ability to engage in sports and physical activities. After 6 months of pain, she had already undergone physiotherapy; taken medications such as anti-inflammatories, muscle relaxants, and 75 mg of pregabalin at night; and participated in postural reorientation training, all without improvement. The patient had a good weight distribution and eating and bowel habits, but reported slightly poor sleep, which she associated with her pain. She was also referred to therapy to address her anxiety.

During a physical examination, a positive Lasègue sign was noted at 30 degrees of MID elevation, with preserved reflexes but slightly decreased strength in the right leg (grade IV in the L5 and S1 myotomes). Paravertebral low back pain was present in the L4-5 and L5-S1 areas, with pain worsening during the forced flexion of the spine, sitting, and with the Valsalva maneuver. Relief was reported with stretching and static standing. The segment degeneration was classified as Pfirmann IV, and the hernia was classified by the Michigan State University (MSU) system as 3B (Figure 1A,B). MRI revealed a right paramedian extruded disc herniation at L5-S1 with the dural and descending root compression of S1, along with segmental degeneration, an arthrosynovial cyst, and right isthmian lysis (Figure 1C, D). Although surgery was recommended by other colleagues, the patient was highly reluctant to undergo any invasive procedure, including minimally invasive or endoscopic surgery, due to severe anxiety and an uncontrollable fear of needles. She was already receiving psychiatric care but remained unsuitable for interventions involving cannulas or needles.

Given these limitations, and with her consent, we opted to apply transcutaneous STP radiofrequency as a therapeutic alternative. Medium adhesive plates were placed on the lumbar spine at the level of the hernia and along the sciatic path in the right buttock, where the pain radiated. A Spring2^®^ generator (from Springlife Medical, Driebergen, The Netherlands) with medium patches was used, and the PRF STP was set to 1.4 A for 15 min, generating 108 V (Figure 1E).

The procedure was repeated at 15 days and 6 weeks after the initial application, following intervals based on the therapeutic response, as described by Sluijter et al. in 2023 [13], for a total of three sessions. The patient continued taking pregabalin as previously prescribed, but after the final session, the medication was discontinued when she reported being completely asymptomatic, with a VAS score of 0. Her right leg strength returned to normal (grade V), and she was able to gradually resume her favorite sport, beach tennis, which involves a significant mechanical impact. At a follow-up consultation 30 days after the final session, she also reported an improved mood and reduced anxiety, attributing part of these symptoms to her inability to engage in her favorite sport.

After 24 months, the patient returned to our clinic with a mild recurrence of pain, localized to the gluteal region without any radiation. Regardless, she still continued to engage in her usual sports activities without claudication or limitations. A new MRI (Figure 2) revealed a significant improvement compared to the previous scan, showing the L5-S1 disc in better condition, though some early signs of degeneration were present in the L4-5 intervertebral disc. While this type of imaging progression is expected due to the natural course of the disease, it is noteworthy that she remained symptom-free and clinically stable for an extended period.

## 3. CASE 2: Cervical Spine Degenerative Disease

Patient 2, a 44-year-old male, was referred to our clinic in São Paulo, Brazil, with complaints of cervical pain radiating to the left arm, accompanied by paresthesias and a worsening pain at night. He reported a VAS score of eight and numbness in his hands, which made it difficult to sleep and perform his work, as he requires precise hand movements for ophthalmic procedures. He had already undergone physiotherapy and used medications such as anti-inflammatories, painkillers, and pregabalin, all without an improvement over three months.

He presented his MRI images (Figure 3A,B). Since he was unable to take time off work for invasive treatments and desired an immediate solution, we suggested Transcutaneous Randomized Pulsed Radiofrequency (STP). After obtaining informed consent, we treated him with three sessions using two small patches for 15 min each, set to 0.8 A and 45 V. The second session took place 15 days after the first, and the final session occurred 45 days after the initial application.

Thirty days after the last session, the patient returned with no pain (VAS 0) and no numbness, which had been his primary complaint. He reported sleeping well and being able to work without difficulty. At a follow-up visit 90 days later, he remained symptom-free and had no further complaints.

## 4. CASE 3: Lumbar Disc with Compression and Central Stenosis

Patient 3, a 32-year-old woman, was referred to our clinic in São Paulo, Brazil, with complaints of low back pain radiating to the left lower limb, accompanied by numbness, which promoted a difficulty sitting for extended periods at work. Her MRI revealed a large disc herniation at L4-L5, compressing the central canal (Figure 4A,B). After 30 days of conventional conservative treatment without improvement, we offered, with her consent, the application of transcutaneous STP pulsed radiofrequency as a therapeutic option. The treatment involved placing medium adhesive patches at the hernia level and along the sciatic path in the left buttock, where the pain was radiating. We used a Spring2^®^ generator with medium patches, set at 1.4 A for 15 min, delivering 108 V of STP (Figure 4C).

The same procedure was repeated in 15 days and the third application was performed 30 days after the second, maintaining intervals based on the therapeutic response as previously described by Sluijter et al. in 2023 [13], resulting in a total of three sessions. By the third visit for the final session, the patient was completely pain-free, with a VAS score of 0, no numbness, and had fully regained her ability to sit at work normally. At a follow-up evaluation 60 days later, she remained pain-free and continued to demonstrate excellent functional outcomes.

## 5. Results

In this case series, we observed positive outcomes, particularly in terms of pain reduction or complete pain relief, with patients becoming asymptomatic within a short follow-up period, ranging from 60 days to 24 months. MRI images showed encouraging improvements after three sessions of transcutaneous PRF. In this small cohort, only one patient experienced a recurrence after 24 months, presenting with mild gluteal pain but without any functional limitations and continuing normal activities.

## 6. Discussion

It is important to emphasize that, as with most medical procedures, these treatments do not offer guaranteed results. Instead, our objective is to offer patients a non-invasive, low-risk opportunity to alleviate symptoms, with reduced hospital demands. Should complaints or clinical signs persist, other options remain available, including invasive techniques using cannulas, endoscopic spine surgery, or, in certain cases, larger surgical interventions such as decompression and fusion.

It is worth noting that all patients in this study completed informed consent forms in accordance with ethical guidelines for non-invasive techniques, acknowledging the minimal risks associated with these procedures. These treatments were chosen because patients did not wish to undergo invasive procedures at the time, and their clinical conditions allowed for such an attempt.

While pulsed radiofrequency is supported by numerous studies showing positive outcomes lasting 6 to 8 months in various spinal disorders, there is limited experience with its transcutaneous application, particularly in its randomized form, known as STP. This technique lacks broad sample sizes with long-term follow-up, and further studies are needed to expand the evidence and produce more robust scientific results.

According to clinical and experimental observations, the developers noticed that normally the effects of pulsed radiofrequency occur in four phases described by Sluijter. The initial phase, called the “Stunning phase”, involves a remarkable improvement that usually lasts 2 to 3 days, followed by a phase of discomfort, where symptoms can reappear, lasting about 3 weeks; the third is the effective phase showing the positive results that can remain up to 6 to 8 months or even years. After that last phase, a recurrence of pain and symptoms may occur; however, some patients had a positive response without recurrence [13].

Using transcutaneous electrodes, we generally follow the Spring2 user manual to determine the size of the paths and the distance between them; small areas indicate the use of small-sized patches (5.5 × 5.5 cm), areas with 12.7 to 24.2 cm of distance between the electrodes indicate the use of medium electrodes (6 × 12 cm), and large areas with a 20 to 36 cm distance are recommended for the use of large patches (8 × 15 cm). And we observed that more sessions would be needed to obtain similar results compared to the conventional invasive pulsed radiofrequency technique. In our practice, after a few pilot cases, we have determined the best way to use it for spinal conditions, showing better results with at least three sessions, as demonstrated here in this case report, but in the future, new possible ways of applying it could be developed.

This technique uses a modulated electromagnetic field, pulsed, and is not continuous; the STP mode uses a modulated sinusoidal wave, square. It uses low frequency rates (3 hz), and the STP mode randomizes it, with all other parameters controlled (pulse width and energy voltage), so it is not involved with high temperature levels, never exceeding 39 degrees Celsius. The temperature here is not the focus of the treatment; it is much more about inflammatory modulation, pain signaling control, and the immune response, as described in the introduction.

The procedure is safe, and we do not have major risks described in a large number of articles written since 1998, when Professor Menno Sluijter created the mode of action of pulsed radiofrequency, as it uses low frequency rates and is not associated with high temperatures. There are risks much more associated with the cannula puncture than with the action of pulsed radiofrequency, so when we use the transcutaneous type, we are able to avoid most of the risks associated with the technique.

It is also understood that many pathologies can be resolved with conservative treatment, and some imaging changes may occur naturally over time. Physiotherapy techniques, such as osteopathic treatments, are expected to produce good results. The basic physiological premise of cranio-sacral therapy is the mobilization of the meninges and the spinal cord through the traction of the cervical spine, the compression of the cranial bones, and reflex treatments to improve the blood supply to the brain, which may also affect the flow of cerebrospinal fluid, aiming for a clinical improvement [28,29]. However, in these cases, all the patients first underwent conservative treatments and used indicated medications, such as analgesics, anti-inflammatories, and pregabalins, insisted on physiotherapy treatments with manual manipulations and TENS (transcutaneous electrical nerve stimulation), and were unable to get rid of the pain or fully recover from the symptoms [25,26]. Of course, we do not have full control over how these treatments were carried out and whether levels of excellence were achieved in all cases, since each patient had their physiotherapy treatment conducted by their physiotherapist of choice, and this may be a bias in the treatments carried out.

However, it is rare to see a sustained clinical improvement in severe degenerative disc disease, as demonstrated in these cases, without resorting to more invasive therapies and sometimes even in indications for spinal surgery. A recent article showed good results with the use of transcutaneous electrical nerve stimulations for acute postoperative pain after spinal surgery, but our aim is to use the pulsed transcutaneous electromagnetic field to try to avoid surgery or other invasive procedures [30].

Pulsed radiofrequency and STP aim to offer a treatment option that targets joints and nerve endings by stimulating inflammation control and redox effects, rather than merely blocking pain through neural lesions. By incorporating these techniques, we can enhance existing therapies and combine treatments to maximize results, with the hope of presenting innovative therapeutic solutions for many chronic pathologies linked to the unhealthy conditions of modern life.

## 7. Conclusions

Transcutaneous Randomized (STP) Pulsed Radiofrequency presents a promising adjuvant tool for the treatment of degenerative spine conditions, even in cases with large disc herniations. This approach has demonstrated a pain reduction and improved mobility in this small cohort of patients. However, traditional surgical interventions remain the preferred option in cases involving sensory or motor neurological deficits or prolonged, persistent symptoms.

## Figures and Tables

**Figure 1 jfmk-10-00242-f001:**
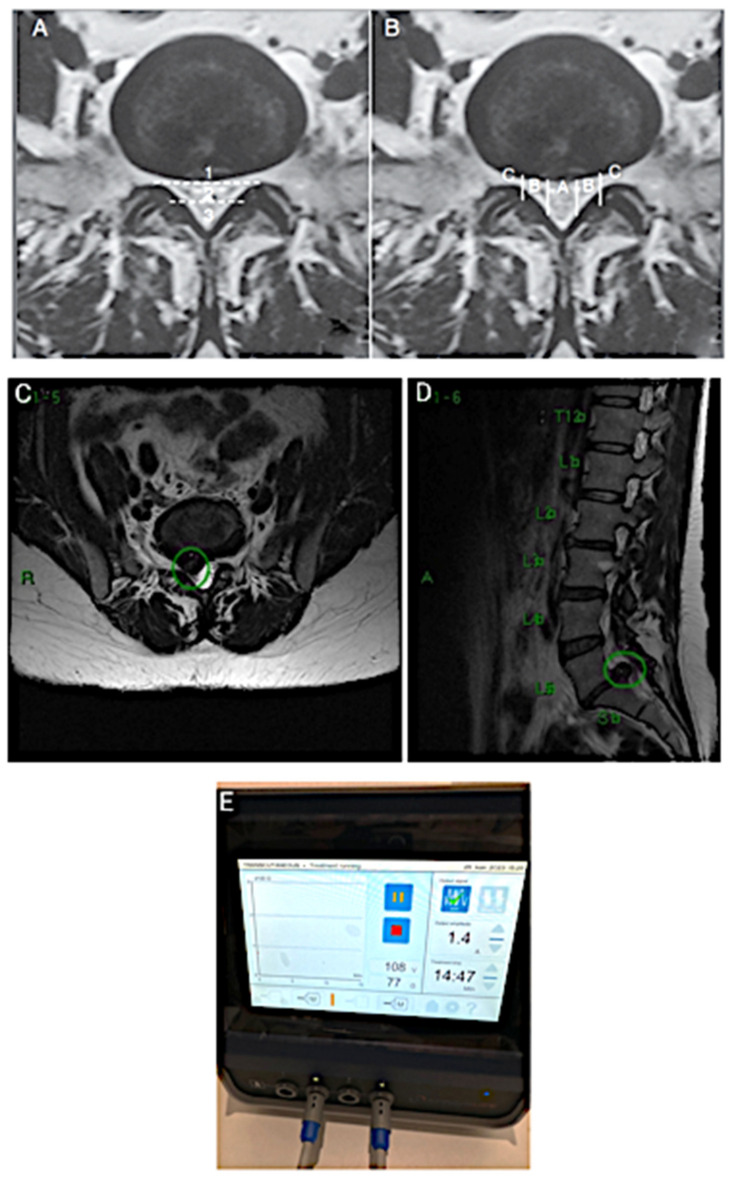
(**A**,**B**) The classification of herniated discs described by MSU (Michigan State University) where numbers 1,2, and 3 describes the disc herniation from more anterior (1) to more posterior into the central spinal canal (3), and the letters ABC describes central disc herniation as A, center–lateral as B, and foraminal or lateral as C; (**C**) Nuclear Magnetic Resonance image demonstrating, at the green circle, a right paramedian L5-S1 extruding hernia with the compression of the dura and contact at the root of S1 on the right; (**D**) Disc herniation taking the entire spinal canal at the green circle, rated as 3B (MSU); and (**E**) an example of TCPRF STP Generator.

**Figure 2 jfmk-10-00242-f002:**
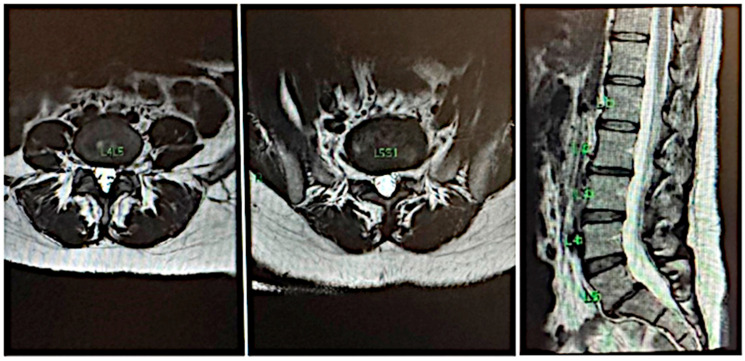
Recent Nuclear Magnetic Resonance Imaging demonstrating good evolution of herniated disc L5-S1.

**Figure 3 jfmk-10-00242-f003:**
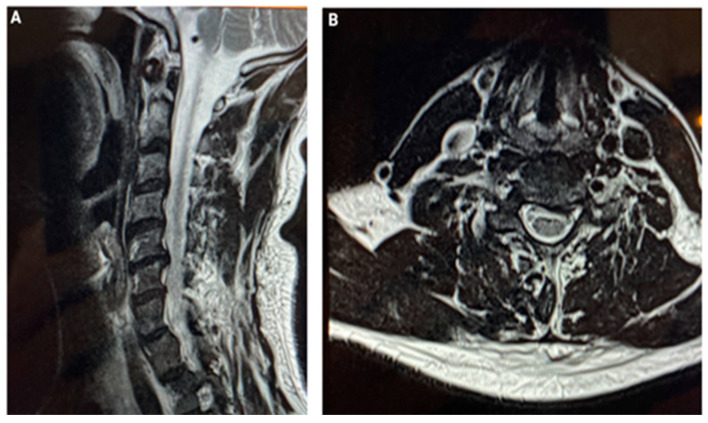
(**A**) A Nuclear Magnetic Resonance image demonstrating the left paramedian C5–C6 disc herniation with contact at the left nerve root. (**B**) The degeneration of the segment and C3–C4 bulging disc.

**Figure 4 jfmk-10-00242-f004:**
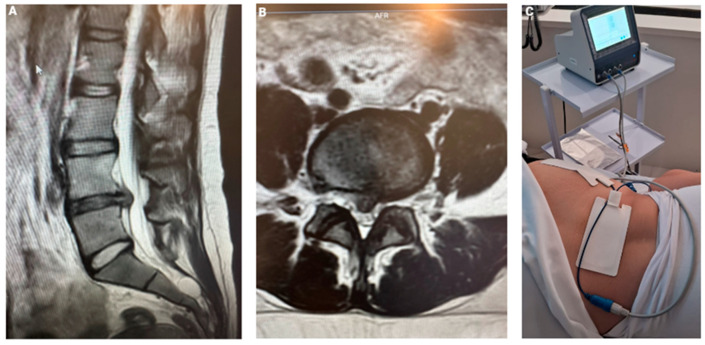
(**A**) The MRI revealed a large disc herniation at L4–L5; (**B**) central canal compression; and (**C**) TC-PRF patch application for lumbar disc herniation with left limb irradiation.

## Data Availability

The original contributions presented in this study are included in the article. Further inquiries can be directed to the corresponding author(s).

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
