# Peer review of "A Transcutaneous Randomized Pulsed Radiofrequency Application for Spine Pain Conditions: A Case Series"

_jfmk, 2025, doi:10.3390/jfmk10030242_

Round 1
Reviewer 1 Report
Comments and Suggestions for Authors
Introduction
The Introduction is too extensive in relation to the rest of the work, especially the Discussion. The Introduction needs to be shortened.
The introduction lacks the purpose of the study.
Case presentation
The case descriptions are done correctly, could they be supplemented with information on whether other treatments or physiotherapy exercises were performed?
Discussion
The discussion is very short. Authors must extensively revise this section. They present a brief summary, but do not confront them with research by other authors. This chapter should be supplemented with reports by other authors. The discussion should be heavily rewritten and supplemented, giving it a scientific character.
Line 301 should be supplemented with information on the possibilities of conservative treatment. Please discuss these issues in the introductory part by referring to the following articles:
- Dwornik, M.; Puszczalowska-Lizis, E.; Wojcik, M.; Szajkowski, S.; Graczykowski, M.; Szymanski, D.; Marszalek, S. Efficacy of osteopathic manipulative treatment (93.6. ICD–9) - systematic review. Medical Studies. 2024; 40 (3): : 289-307. doi: https://doi.org/10.5114/ms.2024.138016.
and other publications dealing with similar issues.
When the authors address these issues I will be able to comment definitively and make the final decision.
Author Response
Thank you very much for your excellent comments! I'll try to answer all the questions below and include them in the revised text
comment 1 ) The Introduction is too extensive in relation to the rest of the work, especially the Discussion. The Introduction needs to be shortened.
response 1 ) The introduction has been shortened and reformulated, introducing the demands made by your comments and those of other reviewers.
comment 2) The case descriptions are done correctly, could they be supplemented with information on whether other treatments or physiotherapy exercises were performed?
response 2) As the patients in question underwent different conservative therapies by their physiotherapists, I have included these descriptions and discussions in the discussion of the article, as we do not have details of these therapeutic actions carried out individually.
comment 3) The discussion is very short. Authors must extensively revise this section. They present a brief summary, but do not confront them with research by other authors. This chapter should be supplemented with reports by other authors. The discussion should be heavily rewritten and supplemented, giving it a scientific character.
response 3) The discussion was broadened to include articles on physiotherapy techniques, opening up such discussions, even though this is not our area of expertise, including the suggested article and others in the field.
comment 4) Please discuss these issues in the introductory part by referring to the following articles:
- Dwornik, M.; Puszczalowska-Lizis, E.; Wojcik, M.; Szajkowski, S.; Graczykowski, M.; Szymanski, D.; Marszalek, S. Efficacy of osteopathic manipulative treatment (93.6. ICD–9) - systematic review. Medical Studies. 2024; 40 (3): : 289-307. doi: https://doi.org/10.5114/ms.2024.138016.
response 4) I have included this article that you suggest and others related to physiotherapy techniques.
Reviewer 2 Report
Comments and Suggestions for Authors
The authors conducted a study on randomized pulsed radiofrequency application for spine pain treatment. The case report is interesting; however, the authors need to clarify a few technical questions. The reviewer’s comment is listed below.
- What’s the rationale for using different RF exposure for all the cases mentioned in this study.
- PRF was used with different types of patches for different patients. What’s the rationale for using different patches for different patients? Explain it.
- What’s the rationale used for determining the number of treatment sessions required for each patient. What will be implication is the patient is treated 2 times instead of 3 times or greater than 3 times.
- Did authors use the same RF setting for all the three sessions? Does patient benefit from applying different RF exposure for different treatment sessions? Could this be a benefit for the patient to reduce the number of treatment sessions.
- Is the RF exposure continuous or modulated? What’s the target tissue temperature during the 15 min treatment? Any elevation from the baseline tissue temperature.
- How is safety assessed during the treatment?
- Does this approach will be more beneficial if there is a temperature elevation to 38 or 39°C?
- Will there be any benefit if the number of sessions were reduced to 1 with a longer RF exposure time. For example, instead of 15 mins what will be the effect for 30 min, 45 min and 1 hr treatment.
Author Response
Answering the questions: Thank you very much for your great questions and mindfull comments!!!
1.The authors involved in this publication (including myself) have been studying pulsed radiofrequency for years, but it is still a new technology, especially the transcutaneous type. The Spring2 transcutaneous pulsed radiofrequency device comes with a user manual, with suggestions made according to the distance between the electrodes, and we use these parameters to treat each case.
2.The size of the electrodes is chosen according to the size of the patient or the indicated area to be treated; for small areas, we use the small one, for large areas, the large one. (We have electrodes available in sizes S, M, L), also following the user manual.
3.We are mainly using 3 sessions, after our last medical meeting held in Leiden, Netherlands, last year. There, we suggested a Pulsed Radiofrequency Consensus, which will also be published soon, with parameters, number of sessions, and other indications, as in fact, there is still no consensus for the technique.
4.Yes, we used the same parameters for all sessions. We don't have all the answers yet. We are studying and starting to publish, trying to enhance Science and increase our knowledge on how to use this type of electromagnetic field.
5.It is modulated, pulsed, and not continuous, so it is not involved with high temperature levels, never exceeding 39 degrees Celsius. The temperature here is not the focus of the treatment; it is much more about inflammatory modulation, pain signaling control, and immune response, as described in the introduction.
6.The procedure is safe, we don't have major risks described in a large number of articles written since 1998, when Professor Menno Sluijter created the mode of action of pulsed radiofrequency, as it uses low frequency rates and is not associated with high temperature. There are risks much more associated with the cannula puncture than with the action of pulsed radiofrequency, so when we use the transcutaneous type, we are far from most risks associated with the technique.
7.Temperature is not the goal of this type of electromagnetic field action.
8.The manufacturer Spring2 does not recommend using it for more than 20 minutes, and the user manual suggests always using 15 minutes for treatments, and we are doing so. But we are starting a new research, where we will investigate all these parameters and the number of sessions in the best way. This is just the first Case Report about the technique, but we are eager to bring much more information in the near future.
Reviewer 3 Report
Comments and Suggestions for Authors
Before reading this manuscript, I had never read about PRF-STP, so I am not well prepared to comment on the extensive review in the Introduction. This review takes about half the length of the text and is followed by the 4 case studies.
The method appears to be astonishing in that it causes rapid reduction in chronic pain and structural recovery of the spine without surgical intervention. This is very remarkable. What is most extraordinary is the correction of the herniated discs. The idea that stimulating currents can block nerves is familiar, and I can readily accept that cytokines are modified, but it seems amazing that orthopaedic injuries can be corrected so quickly without physical intervention. However, it seems to have worked on the four patients presented in the second half of the paper, whether the effect is understood or not.
I suggest that the paper should be improved in two ways.
First, the stimulation should be fully described. Is it a controlled-current stimulator? Is the radio-frequency 420kHz? Is this sine, square or other waveform? Are the bursts on/off or with analogue modulation? How would the irregular intervals between bursts be described? How large are the self-adhesive electrodes? What does the patient feel at this intensity? How do you choose the intensity of the current? Why is the dose fifteen minutes?
Second, sceptical readers may suspect that you are only presenting your successes but not your failures. Did you use this method on any other patients which you have not included in this paper? If you did not, then I think it is particularly important to say so, because it strengthens your case that this appears to be a low-cost, effective treatment for people disabled by serious pain.
Author Response
Answering the questions: Thank you very much for your great questions and attentive comments!!!
The authors involved in this publication have been studying pulsed radiofrequency for years, but it is still a new technology, especially the transcutaneous type. The Spring2 transcutaneous pulsed radiofrequency device comes with a user manual, with suggestions made according to the distance between the electrodes and the diameter of the limb to be treated, and we use these parameters to treat each case.
The size of the electrodes is chosen according to the size of the patient or the indicated area to be treated; for small areas, we use the small one, for large areas, the large one. (We have electrodes available in sizes S, M, L), also following the user manual.
We are studying and starting to publish, trying to enhance Science and increase our knowledge on how to use this type of electromagnetic field.
It is modulated eletromagnetic field, pulsed, and not continuous, the STP mode uses a modulated sinusoidal wave, square. it uses low frequency rates ( 3hz ) and the STP mode randomize it between 2 to 8 Hz with all other parameters controled ( pulse width and energy Voltage ) so it is not involved with high temperature levels, never exceeding 39 degrees Celsius. The temperature here is not the focus of the treatment; it is much more about inflammatory modulation, pain signaling control, and immune response, as described in the introduction.
The procedure is safe, we don't have major risks described in a large number of articles written since 1998, when Professor Menno Sluijter created the mode of action of pulsed radiofrequency, as it uses low frequency rates and is not associated with high temperature. There are risks much more associated with the cannula puncture than with the action of pulsed radiofrequency, so when we use the transcutaneous type, we are far from most risks associated with the technique.
The manufacturer Spring2 does not recommend using it for more than 20 minutes, and the user manual suggests always using 15 minutes for treatments, and we are doing so. But we are starting a new research, where we will investigate all these parameters and the number of sessions in the best way.
This is just the first Case Report about the technique, but we are eager to bring much more information in the near future. Unfortunately, one participant no longer wants to participate or be exposed in the work, and I will need to remove case 2 from the report, but we will proceed with the others. I do have some new cases ongoing, most with good results, but a few with weak results; all of these cases will be published in the future. I will also revise the text with the relevant information you suggested! Thank you very much for your amazing contribution!!!
Round 2
Reviewer 1 Report
Comments and Suggestions for Authors I accept the manuscript in its current form.Reviewer 2 Report
Comments and Suggestions for Authors
The authors responded well to the questions raised by the reviewer and recommends the publication of the article